# Quasi-BIC-Based High-Q Perfect Absorber with Decoupled Resonant Wavelength and Q Factor

Weiyi Zha [1], Yun Huang [1], Pintu Ghosh [1,2,*] and Qiang Li [1]

1   State Key Laboratory of Modern Optical Instrumentation, College of Optical Science and Engineering, Zhejiang University, Hangzhou 310024, China; 11830049@zju.edu.cn (W.Z.); yun_huang@zju.edu.cn (Y.H.); qiangli@zju.edu.cn (Q.L.)
2   Department of Physics, Medi-Caps University, Indore 453331, India
*   Correspondence: pintuiitb@gmail.com

**Abstract:** The Q factor in a quasi-BIC-based optical device can approach infinity and has therefore been attracting the attention of many researchers in recent years. However, this mode is barely applied to absorbers since it mainly tunes the radiative loss. The resonant wavelength of quasi-BICs normally couples with the Q factor, and it is difficult to independently tune one of them while maintaining the other, which weakens the flexibility of tuning. In this work, a quasi-BIC-based high-Q perfect absorber with some unique features is proposed. It shows a decoupled relationship between the resonant wavelength and the Q factor such that these two properties can be independently tuned by changing different structure parameters. In addition, both radiative and resistive losses are tunable. An easy method is proposed to design a perfect absorber with different resonant wavelengths and different Q factors, and a near-infrared perfect absorber with a Q factor as high as $5.13 \times 10^5$ is designed. This work proposes a method to tune the quasi-BIC mode, thereby introducing a new paradigm for the design of a high-Q perfect absorber.

**Keywords:** quasi-BIC; high Q; perfect absorber; decoupled resonant wavelength and Q factor

## 1. Introduction

A high-Q factor means a long photon lifetime and good coherence, which suggests high accuracy and high quality in practical applications. High-Q devices have been attracting the attention of many researchers for decades and are extensively applied in the fields of sensing [1,2], lasing [3], filtering [4], information transfer [5], nonlinear harmonics generations [6], energy manipulation [7], and so on. High-Q absorbers can be used in sensors [8] and monochromatic photoelectric detectors [9]. In the infrared range, high-Q absorbers can also work as a high-Q infrared source [10] since absorptivity is equal to emissivity according to Kirchhoff's law of thermal radiation. High-Q absorbers can be realized by using photonic crystals [11], Tamm plasmonic structures [12], surface plasmon polariton (SPP) [13], surface phonon polariton (SPhP) [14], Bragg gratings [15], and so on.

Bound states in the continuum (BICs) [16,17] are special electromagnetic states that are confined in photonic structures and cannot radiate to the far field, even though they coexist with a continuous spectrum that can carry energy away. They can exist in various photonic structures, e.g., metasurfaces [18,19], plasmonic structures [20,21], photonic crystals [22,23], and fiber Bragg gratings [24]. BICs can be generally divided into three categories—symmetry-protected BICs [25,26], single-resonance parametric BICs [27], and coupled resonance induced BICs [28,29]. The symmetry-protected BICs exist in a system exhibiting a reflection or rotational symmetry. The single-resonance parametric BICs can be evolved from a single resonance when enough parameters are tuned. The coupled resonance induced BICs originate from the coupling between two or more resonances in the system.

Quasi-BICs are often evolved from coupled resonance induced BICs (e.g., Fano resonances [30,31] and Mie resonances [32,33]), where the infinite radiative Q degrades to a finite value. In most works, the resonances forming quasi-BICs share the same optical cavity and are simultaneously tuned when changing the structure parameters [34,35], so the resonant wavelength and the Q factor change together during the process; i.e., the resonant wavelength is coupled with the Q factor in the quasi-BIC system, which weakens the flexibility of tuning. Since they are mainly used to tune radiative loss, current quasi-BICs are often applied in reflectors or transmitters [36,37] and barely in absorbers.

In this work, a quasi-BIC-based high-Q perfect absorber is proposed by using the coupling between a guided mode and a Fabry–Perot (FP) mode. The resonant wavelength and the Q factor of the quasi-BIC mode are decoupled and can be tuned independently. Both resistive and radiative losses can be tuned in the structure. An easy method to design a perfect absorber with different resonant wavelengths and different Q factors is proposed. A near-infrared perfect absorber with a Q factor as high as $5.13 \times 10^5$ is designed by using this method.

## 2. Methods

Simulations are conducted in commercial software COMSOL 5.4.

Simulation of absorptivity: Cross-section of a single period was built in COMSOL 5.4, with air layers upon the structure. "Electromagnetic Waves, Frequency Domain" and "Frequency Domain" were chosen for "Physics" and "Study" parts, respectively. Periodic boundary condition was set for lateral boundaries. The top air layer was set as a perfectly matched layer (PML). The bottom boundary of the PML was set as a port to emit light to the structure. After setting the material parameters (relative permittivity, relative permeability, and electrical conductivity), meshing, and scanning parameters, calculations were conducted. Finally, reflectivity $R$ can be calculated by taking square of S-parameter $S_{11}$, and absorptivity can be obtained by $1-R$ since there is no transmission in the system.

Simulation of eigenmode: Model building, boundary conditions, and "Physics" were the same as the simulation of absorptivity above, except that there was no need of a port. "Eigenfrequency" was chosen for the "Study" part. For the simulation of radiative loss, the imaginary part of relative permittivity should be set as 0, and in this case, the energy loss in the system is the radiative loss. For the simulation of resistive loss, the imaginary part of relative permittivity maintains the original, and the energy loss in the system is the total loss (the sum of resistive and radiative losses). Then, the resistive loss can be evaluated from these simulations.

## 3. Results and Discussion

### 3.1. Scheme of the Quasi-BIC-Based High-Q Perfect Absorber

The quasi-BIC-based high-Q absorber in this work is composed of a Au substrate, a $SiO_2$ spacer, and a Si layer from bottom to top, with gratings engraved on top of the Si layer (left panel in Figure 1). In this structure, BIC originated from the coupling between a guided mode and an FP mode. The guided mode horizontally propagates inside the Si layer (since the refractive index of Si is higher than that of $SiO_2$), while the FP mode vertically oscillates between the Si and $SiO_2$ layers (right panel in Figure 1). The Au substrate serves as a bottom mirror to prevent transmission and absorb the incident light. The $SiO_2$ layer is used to separate the guided mode from the Au substrate to reduce the resistive loss. The resonance of the guided mode is determined by the period of gratings and the thickness of the Si layer, while that of the FP mode is determined by the thickness of both Si and $SiO_2$ layers. Since the factors affecting the resonance of both modes slightly differ from each other, it is possible to tune one mode while maintaining the other. Thus, it is feasible to independently tune the resonant wavelength and the Q factor of the quasi-BIC mode. The cross-section of a single period is demonstrated in the middle panel of Figure 1, where $w$ represents the width of the grating; $p$ is the period; $h_0$ is the grating thickness; $h_1$ is the Si layer thickness; $h_2$ is the $SiO_2$ layer thickness, and $h_3$ is the thickness of the Au substrate.

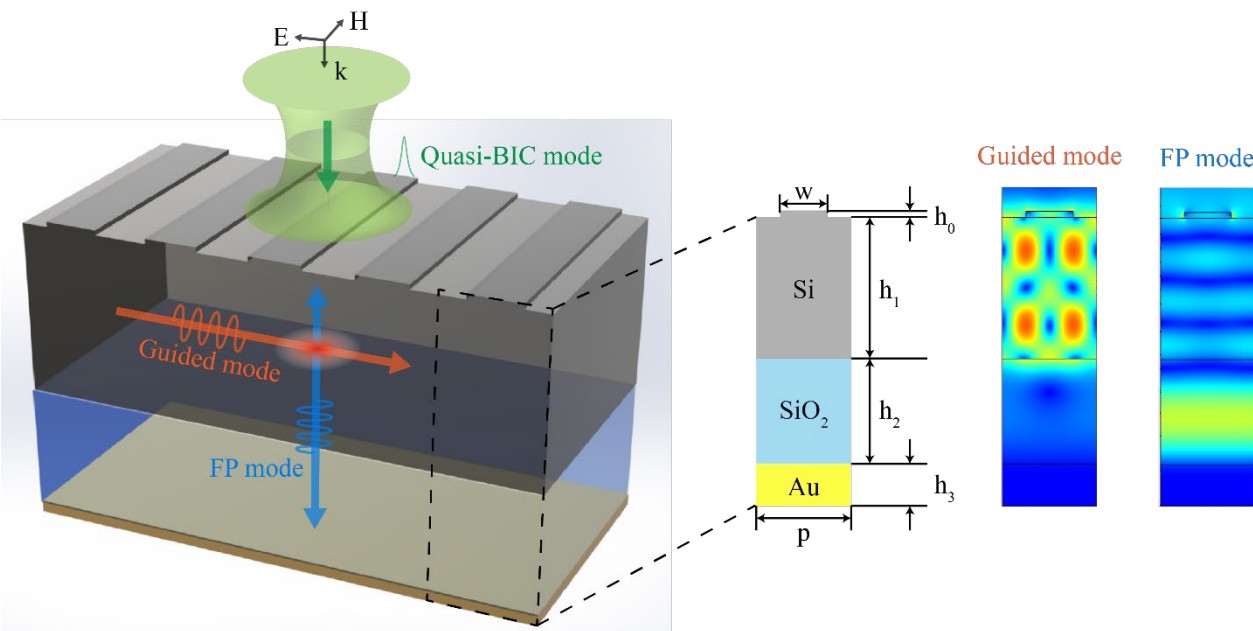

**Figure 1.** Schematic of the quasi-BIC based high-Q perfect absorber (**left panel**), the cross-section of a single period (**middle panel**), and electric field distributions of the guided mode and the FP mode (**right panel**).

*3.2. Characterization of the BIC Mode*

To study the characteristics of the BIC mode illustrated above, the dispersion relationship of absorptivity versus wavelength $\lambda$ and the $SiO_2$ layer thickness $h_2$ was simulated for the structures with different periods $p$ = 430 nm, 450 nm, and 470 nm (Figure 2a–c). Other structure parameters were set as $h_0$ = 30 nm, $h_1$ = 670 nm, $h_3$ = 200 nm, and $w = p/2$. The track of the guided mode is parallel to the $h_2$ axis, while that of the FP mode is almost an oblique straight line (Figure 2a–c). The resonance of the FP mode experiences a redshift with increasing $h_2$, while the guided mode maintains the same resonant wavelength. On the other hand, the track of the FP mode maintains the same position, while that of the guided mode redshifts with increasing $p$. Thus, the resonances of the guided mode and the FP mode can be independently tuned by either changing $p$ or $h_2$.

The BIC point (white point in Figure 2a–c, where the peak absorptivity is zero) occurs in the vicinity of the avoided crossing (red circled area in Figure 2a–c). It is situated at the track of the guided mode and far from the track of the FP mode, and the quasi-BIC modes around the BIC point almost share the same resonant wavelength. The $h_2$ values of each BIC point are about 480 nm, 560 nm, and 660 nm corresponding to $p$ = 430 nm, 450 nm, and 470 nm, respectively. By comparing Figure 2a–c, it is noted that the resonant wavelength of the quasi-BIC mode is relevant to the period $p$ and is hardly influenced by the thickness of $SiO_2$ layer $h_2$.

To study the influence of $p$ and $h_2$ on the quasi-BIC modes, five absorptivity curves (Figure 2d–f) were plotted for the quasi-BIC modes in the orange dashed areas of Figure 2a–c, with the interval of $h_2$ as 10 nm. For all the periods, the difference between the resonant wavelengths is within 1 nm against the change of $h_2$. Especially for $p$ = 470 nm, the quasi-BIC modes resonate at the same wavelength of 1356 nm (Figure 2f), showing that the resonant wavelength is hardly influenced by $h_2$. However, the resonant wavelength changes by nearly 40 nm when $p$ changes by 20 nm. The resonant wavelengths are near 1275 nm, 1312 nm, and 1356 nm for $p$ = 430 nm, 450 nm, and 470 nm, respectively (Figure 2d–f), indicating that the period is a key factor to change the resonant wavelength.

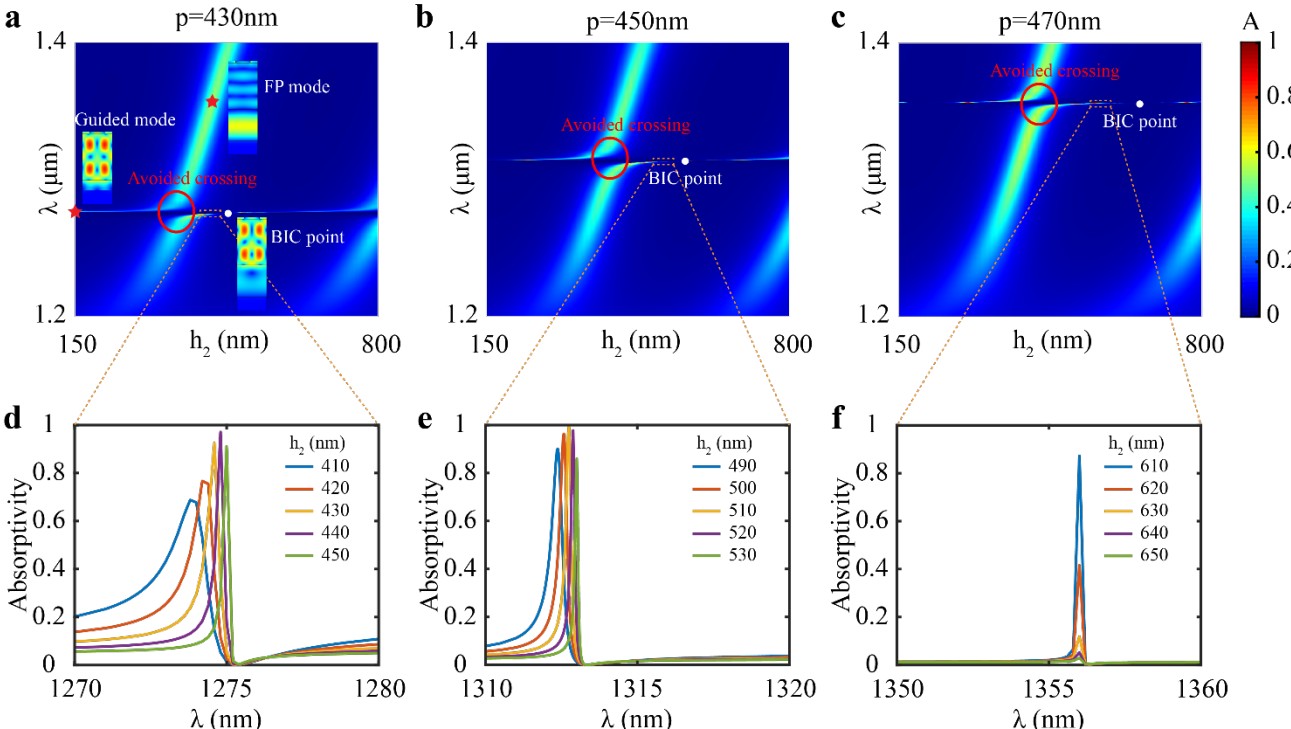

**Figure 2.** Simulation results of absorptivity. (**a–c**) Dispersion relation of absorptivity versus wavelength and the SiO$_2$ layer thickness for the period $p$ values of 430 nm, 450 nm, and 470 nm, respectively. Insets show electric field distributions in (**a**) corresponding to the guided mode and the FP mode (the red stars) and the BIC point (the white point). Red circled area represents the avoided crossing. (**d–f**) The absorptivity curves for different SiO$_2$ layer thicknesses $h_2$ (with the interval of 10 nm), corresponding to the indicated orange dashed area of (**a–c**), respectively.

Besides the resonant wavelength, the Q factor (or the bandwidth) and the peak absorptivity of the quasi-BIC mode were also studied from Figure 2d–f. For $p = 430$ nm, the Q factor becomes higher (the bandwidth becomes narrower) when $h_2$ is tuned from 410 nm to 450 nm (Figure 2d); for $p = 470$ nm, the peak absorptivity decreases when $h_2$ is tuned from 610 nm to 650 nm (Figure 2f). These results suggest that $h_2$ cannot be used to tune the resonant wavelength, but it can help to tune the Q factor (or the bandwidth) and the peak absorptivity.

These simulation results indicate that the resonant wavelength and the Q factor (the peak absorptivity) can be independently tuned in the quasi-BIC mode by changing either the period $p$ or the SiO$_2$ layer thickness $h_2$.

### 3.3. Analysis of Q Factor and Peak Absorptivity

To further study the mechanism behind the Q factor and the peak absorptivity of the quasi-BIC mode, the energy damping rate was simulated by using the eigenfrequency method. Different parameters ($p$, $h_2$, $h_0 + h_1$, etc.) influence the coupling between the guided mode and the FP mode in a similar way, i.e., by tuning the phase difference between these two modes. Since the period $p$ was used to determine the resonant wavelength and the SiO$_2$ layer thickness $h_2$ was used to tune the coupling, $h_2$ was set as the only variable in the following simulation, with other structure parameters as $h_0 = 30$ nm, $h_1 = 670$ nm, $h_3 = 200$ nm, $p = 450$ nm, and $w = 225$ nm (Figure 3a). When $h_2$ is changed from 510 nm to 600 nm (the BIC point is within this range), the damping rate of resistive loss maintains the magnitude of $10^{10}$ s$^{-1}$, while that of radiative loss experiences a huge change from $10^7$ s$^{-1}$ to $10^{10}$ s$^{-1}$ (Figure 3b). This indicates that $h_2$ mainly influences the radiative loss and slightly influences the resistive loss. The change of resonant wavelength stays within 1 nm during the tuning of $h_2$ (Figure 3e). For $h_2 = 550$ nm (near the BIC point), the damping

rate of resistive loss is nearly three orders higher than that of the radiative loss. According to the coupled mode theory [38], resistive and radiative losses serve key roles in tuning the Q factor and the peak absorptivity. Thus, the mechanism behind the Q factor and peak absorptivity tuning is that $h_2$ can change the radiative loss.

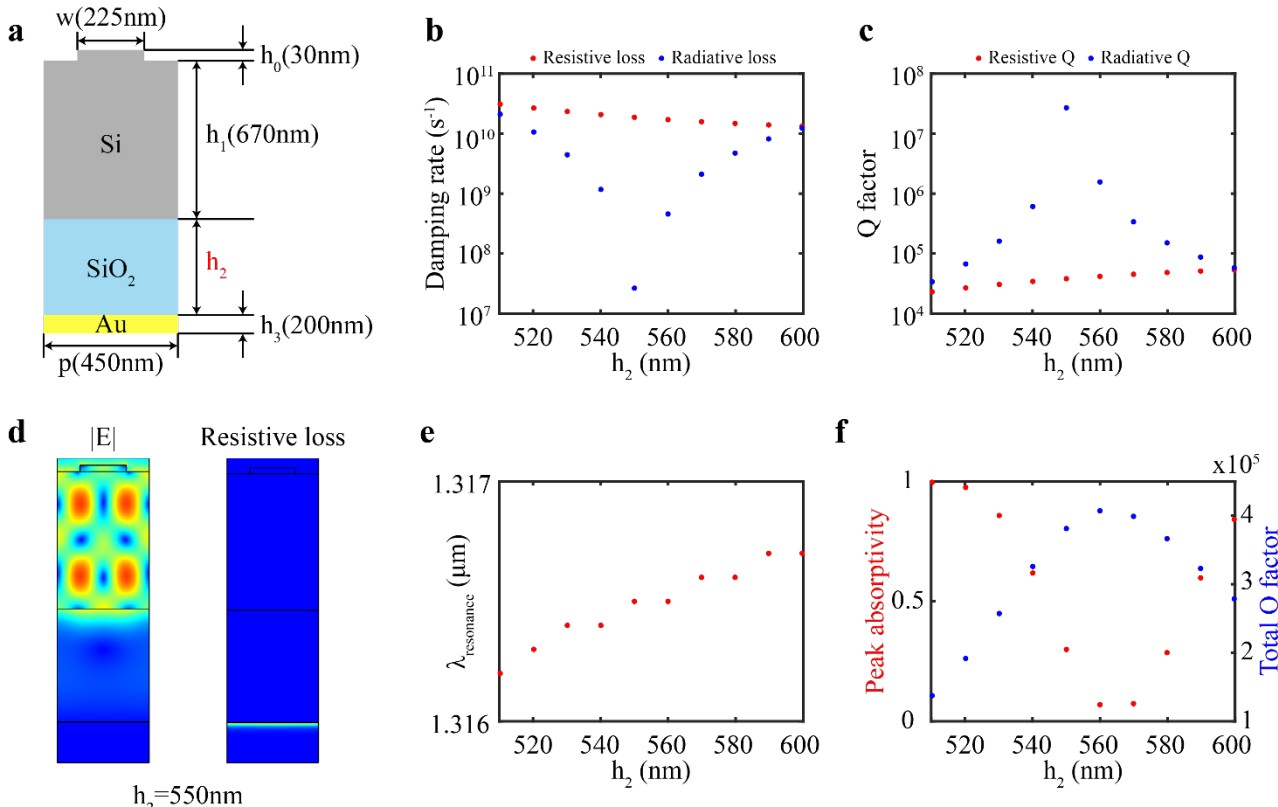

**Figure 3.** Analysis of Q factor and peak absorptivity. (**a**) Schematic of the simulated structure. The only variable parameter is the SiO₂ layer thickness $h_2$. Plots of (**b**) damping rates and (**c**) Q factors versus $h_2$. Red and blue dots represent resistive and radiative losses, respectively. (**d**) Distribution of electric field (left) and resistive loss (right) for $h_2 = 550$ nm. (**e**) The resonant wavelength versus $h_2$ plots. (**f**) Peak absorptivity (red dots) and total Q factor (blue dots) versus $h_2$ plots. For all the above cases, $h_2$ is tuned from 510 nm to 610 nm, with an interval of 10 nm.

The resistive Q factor and radiative Q factor can be transferred from the energy damping rate by using the formula $Q = \frac{\omega}{2\gamma}$ ($\omega$ and $\gamma$ represent the resonant angular frequency and the energy damping rate, respectively), as shown in Figure 3c. The total Q factor of the resonance can be obtained from $\frac{1}{Q_{tot}} = \frac{1}{Q_{res}} + \frac{1}{Q_{rad}}$ ($Q_{tot}$, $Q_{res}$, and $Q_{rad}$ represent the total Q factor, resistive Q factor, and radiative Q factor, respectively). Thus, unlike dielectric BIC structures (without resistive loss), the total Q factor of the BIC mode in this work is limited by the resistive loss and cannot approach infinity. The resistive loss in this structure is mainly from the bottom Au substrate (Figure 3d). Moreover, the dissipation mainly happens within the top 50 nm of the Au substrate, which suggests that as long as the thickness of the Au substrate is larger than 50 nm, there will be no light transmission, and the absorption will be the same.

Comparing the simulation results of Figure 3b,f, it can be concluded that the closer the damping rates of resistive loss and radiative loss, the higher the peak absorptivity (i.e., closer to the condition of critical coupling). Unlike the total Q factor, the peak absorptivity decreases first and then increases during the change of $h_2$ from 510 nm to 600 nm; i.e., the highest total Q factor corresponds to the lowest peak absorptivity. Thus, there is a tradeoff between the total Q factor and the peak absorptivity for this design of the quasi-BIC high-Q absorbers presented in this work.

### 3.4. Realizing Higher Q Factor While Maintaining High Peak Absorptivity

From the analysis above, high peak absorptivity of the quasi-BIC mode requires the damping rates of resistive loss and radiative loss to be as close as possible. Since the resistive loss is hardly influenced by the $SiO_2$ layer thickness $h_2$, one idea to simultaneously realize a higher Q factor and high peak absorptivity in the quasi-BIC mode is to decrease the damping rate of resistive loss at first and then tune that of radiative loss to the same magnitude as the resistive loss by changing $h_2$. Thus, it is critical to find a way to lower the damping rate of resistive loss in the quasi-BIC mode.

The grating thickness $h_0$ is found to have the capability to change the damping rate of resistive loss. Simulations are conducted for the structure with parameters $h_0 + h_1 = 700$ nm, $h_2 = 530$ nm, $h_3 = 200$ nm, $p = 450$ nm, and $w = 225$ nm (Figure 4a). The damping rate of resistive loss increases by two orders—from $10^9$ s$^{-1}$ to $10^{11}$ s$^{-1}$ when $h_0$ is changed from 5 nm to 70 nm (Figure 4b). The damping rate of radiative loss decreases to $10^6$ s$^{-1}$ at first and then increases rapidly to $10^{11}$ s$^{-1}$, suggesting that the resonance evolves to a BIC mode during the progress.

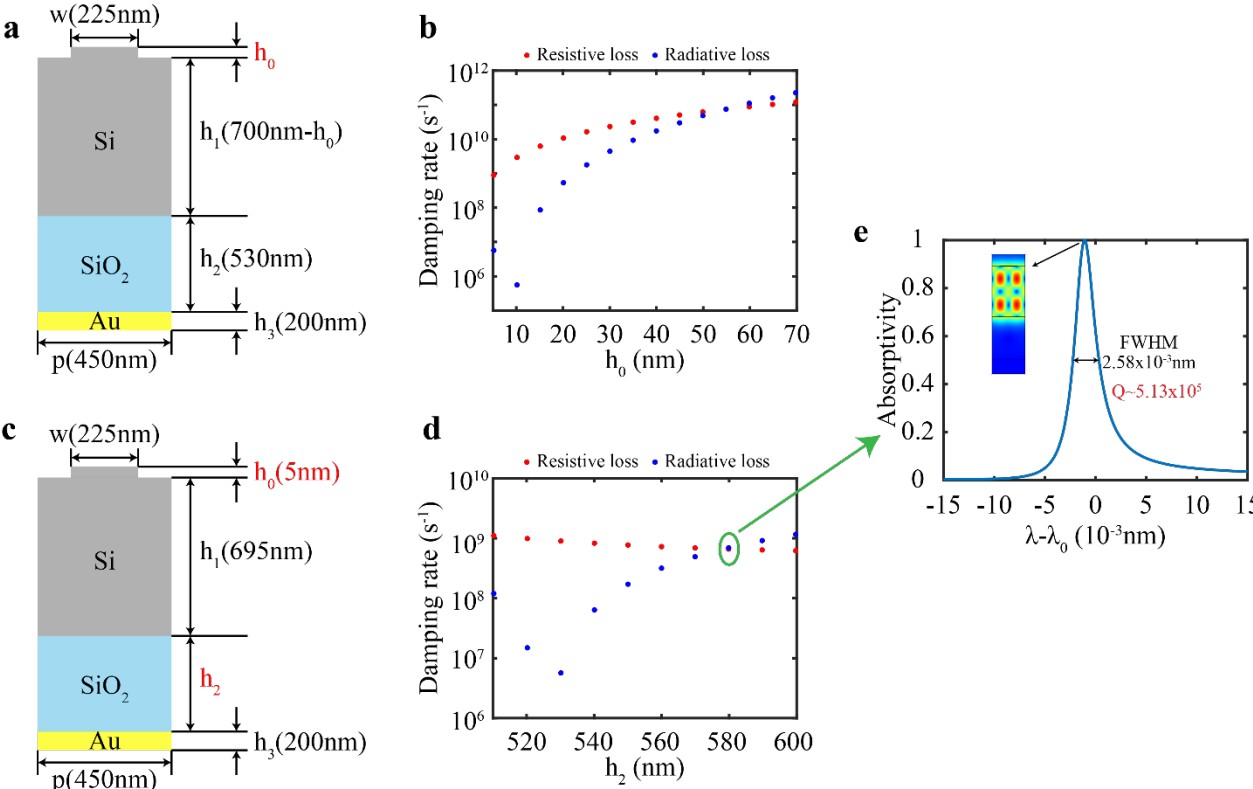

**Figure 4.** Design for increasing Q factor. (**a**) Schematic of the structure (grating thickness $h_0$ is the variable parameter). (**b**) Damping rates versus $h_0$ plots. Red and blue dots represent resistive and radiative losses, respectively. Grating thickness $h_0$ is varied from 5 nm to 70 nm, with an interval of 5 nm. (**c**) Schematic of the structure ($h_0$ is set as 5 nm, and the $SiO_2$ layer thickness $h_2$ is varied). (**d**) Damping rates versus $h_2$ plots. $SiO_2$ layer thickness $h_2$ is changed from 510 nm to 600 nm, with an interval of 10 nm. (**e**) Absorptivity curve for the green circled structure parameter in (**d**). Inset represents the electric field distribution at the peak, $\lambda_0 = 1323.375$ nm.

According to the design idea, $h_0$ was set as 5 nm to decrease the damping rate of resistive loss as much as possible, and the next step was to tune the damping rate of radiative loss to the magnitude of resistive loss (Figure 4c,d). Setting $h_0$ as 5 nm and keeping other structure parameters fixed, damping rates were simulated for different $h_2$ values (from 510 nm to 600 nm). The damping rates of resistive loss and radiative loss almost equal to each other when $h_2 = 580$ nm (green circled data in Figure 4d).

Finally, a quasi-BIC-based high-Q near-infrared perfect absorber was designed with the following structure parameters: $h_0$ = 5 nm, $h_1$ = 695 nm, $h_2$ = 580 nm, $h_3$ = 200 nm, $p$ = 450 nm, and $w$ = 225 nm. The absorptivity curve is shown in Figure 4e. The structure has a peak at 1323.375 nm (the electric field distribution is shown in the inset), and the peak absorptivity is unity. The FWHM (full width at half maximum) of the peak is as narrow as $2.58 \times 10^{-3}$ nm, corresponding to a high Q factor of $5.13 \times 10^5$.

### 3.5. General Steps to Design a Quasi-BIC-Based High-Q Perfect Absorber

Based on the structure in this work, a general idea to design a quasi-BIC-based high-Q absorber with high peak absorptivity can be formed. First, manipulate the resonant wavelength by tuning the period $p$; second, tailor the damping rate of resistive loss by tuning the grating thickness $h_0$; last, make the damping rate of radiative loss approximately equal to that of resistive loss by tuning the SiO$_2$ layer thickness $h_2$.

## 4. Conclusions

We demonstrate a method to decouple the resonant wavelength and the Q factor in a quasi-BIC-based high-Q perfect absorber, and by using this method a near-infrared perfect absorber with Q factor as high as $5.13 \times 10^5$ is designed. First, the design in this work is versatile such that it can be easily scaled to different operating wavelengths, e.g., mid-IR for sensing or high-Q emitter. Second, the mechanism of decoupling the resonant wavelength and the Q factor can used on other structures with two resonances that can be independently tuned. Third, by replacing the SiO$_2$ layer with a dielectric elastomer actuator [39] (thickness could be varied dynamically), it could be possible to dynamically tune the Q factor while maintaining the resonant wavelength. Lastly, the structure proposed in this work can serve as a platform for cutting-edge technologies [40–46], including ultrasensitive biosensing [47], high-harmonic generations [48], and coherent and quantum light generations [49].

**Author Contributions:** W.Z. and Y.H. contributed equally to this work. Conceptualization, W.Z., Y.H., P.G. and Q.L.; methodology, W.Z. and Y.H.; software, W.Z. and Y.H.; validation, W.Z. and Y.H.; formal analysis, W.Z. and Y.H.; investigation, W.Z.; resources, W.Z.; data curation, Y.H.; writing—original draft preparation, Y.H.; writing—review and editing, P.G. and Q.L.; visualization, Y.H.; supervision, P.G. and Q.L.; project administration, Q.L.; funding acquisition, Q.L. All authors have read and agreed to the published version of the manuscript.

**Funding:** This work was supported by the National Natural Science Foundation of China (61975181).

**Conflicts of Interest:** The authors declare no conflict of interest.

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
