# Peer review of "Quasi-BIC-Based High-Q Perfect Absorber with Decoupled Resonant Wavelength and Q Factor"

_electronics, doi:10.3390/electronics11152313_

Round 1

Reviewer 1 Report

Y. Huang et al. have demonstrated a numerical analysis to decouple the resonant wavelength and the Q factor in a quasi-BIC-based high-Q perfect absorber. By varying the structural parameters such as the height of the SiO2 spacer layer and the periods of the grating, they could tune the absorptivity. Besides they have also calculated the resistive and radiative losses of the absorber. They reported a Q factor of 5.13 x 105 for their structure which is also quite close to the recent report (Applied Physics 131, 213104 (2022) of a high Q absorber with a Q factor of 1.7 x 105. How does this approach different in comparison from the recent development reported in Journal of Applied Physics 131, 213104 (2022). How does the absorption vary with the angle of incidence? Overall it is a nice numerical study to calculate the performance of a high Q absorber, though could not successfully demonstrate its significance.

Reviewer 2 Report

The paper is devoted to development of methods and devices to enhance the Q-factor significantly. A device for near-infrared range with Q factor as high as 5.13x105 is designed.

The text is good written, the figures and and literatutre correspond to the content.

The paper can be published as is.

Author Response

We thank the reviewer for the recognition of this work.

Reviewer 3 Report

This submission reports an interesting, but not well organized, demonstration of a method to decouple the resonant wavelength and the Q factor in a quasi-BIC-based high-Q perfect absorber. The quasi-BIC-based high-Q absorber in this work is composed of an Au substrate, a SiO2 spacer, and a Si layer from bottom to top in their theoretical design. Please explain more why authors choose these materials in their design to form the absorbor. While the manuscript contains enough useful information, but more physics mechanism behind simulation results needs to be explained and added in many places in the manuscript.

Some additional concerns should also be addressed:

(1)   In manuscript, there are descriptions that are not clear and needs to be further explained The SiO2 layer thickness h2 is the only variable and other structure parameters are set as h0=30 nm, h1=670 nm, h3=200 nm, p=450 nm, and w=225 nm (Figure 3(a)).’

Why did authors set h2 the only variable and other parameters as those values? Do these parameters follow some principles?

(2)   In section 3.3, authors wrote ‘Thus, unlike dielectric BIC structures (without resistive loss), the total Q factor of the BIC mode in this work is limited by the resistive loss and cannot approach infinity. The resistive loss in this structure is mainly from the bottom Au substrate (Figure 3(d)).’

The thickness of Au substrate may influence the resistive loss. The authors should explain how the thickness of Au substrate can influence the resistive loss and thus influence the absorptivity and Q-factor.

There are other methods and structures that can decouple the resonant wavelength and the Q factor. Please explain the novelty and advantages of the design principles and structure that authors proposed.

Round 2

Reviewer 3 Report

The authors have responded my comments.